# Rational Design of a Polyurethane Foam

**DOI:** 10.3390/polym14235111

**Published:** 2022-11-24

**Authors:** Harry Charles Wright, Duncan Drummond Cameron, Anthony John Ryan

**Affiliations:** 1Department of Chemistry, The University of Sheffield, Sheffield S3 7HF, UK; 2Plants, Photosynthesis and Soil, School of Biosciences, The University of Sheffield, Sheffield S10 2TN, UK

**Keywords:** polyurethane, foam, formulation, experimental design, DoE

## Abstract

Polyurethane (PU) foams are exceptionally versatile due to the nature of PU bond formation and the large variety of polymeric backbones and formulation components such as catalysts and surfactants. This versatility introduces a challenge, namely a near unlimited number of variables for formulating foams. In addition to this, PU foam development requires expert knowledge, not only in polyurethane chemistry but also in the art of evaluating the resulting foams. In this work, we demonstrate that a rational experimental design framework in conjunction with a design of experiments (DoE) approach reduces both the number of experiments required to understand the formulation space and reduces the need for tacit knowledge from a PU expert. We focus on an in-depth example where a catalyst and two surfactants of a known formulation are set as factors and foam physical properties are set as responses. An iterative DoE approach is used to generate a set of foams with substantially different cell morphology and hydrodynamic behaviour. We demonstrate that with 23 screening formulations and 16 final formulations, foam physical properties can be modelled from catalyst and surfactant loadings. This approach also allows for the exploration of relationships between the cell morphology of PU foam and its hydrodynamic behaviour.

## 1. Introduction

Polyurethane (PU) foams are exceptionally versatile due to the nature of PU bond formation, essentially “click” chemistry, with reactions that are “modular, wide in scope and give very high yields” [1]. The large variety of aliphatic, aromatic and polymeric backbones as well as other formulation components such as catalysts and surfactants add to this versatile [2]. Figure 1A shows the quantitative linking reactions between an isocyanate and polyol to form a urethane and Figure 1B shows the quantitative linking reaction between an isocyanate and an amine to form a disubstituted urea where R, R’ and R” represent the plethora of monomers, polymer backbones and functionality that can be utilised in PU foam formulation. Table 1, adapted from Szychers, 2013 [3] shows a generalised formulation for a PU foams and the components that make up such foams. The number of formulation components is often further increased via the use of combinations of polyols, blowing agents, catalysts, surfactants, and additives to obtain application specific properties [4]. Because of their versatility, PU foams are used in a variety of applications ranging from insulating materials and comfort (bedding, furniture, etc.) to shoe insoles or carpet underlay [5].

This versatility, in monomers, polymer backbone chemistry and formulation components, introduces a challenge, namely a near unlimited number of variables for formulating foams. PU foam development requires expert knowledge, not only in polyurethane chemistry but also in the art of evaluating the resulting foams [3,6]. The number of possible formulation variables and the need for both expert knowledge based on years of craft experience means that generating a hard set of rules for the formulation of PU foams is impossible [7]. A more efficient approach is the use of a semi-structured and rational experimental design framework in conjunction with a design of experiments (DoE) approach to overcome some of these challenges; by reducing the number of experiments required to understand the formulation space and foam components. DoE approaches have shown useful in PU foam formulation where DoE has been used to: develop structure-property relationships of foams with varying polyols [8], optimised biobased PU foams with wheat straw derived polyols [9], optimised additives and additive loading when designing foams for acoustical applications [10] and optimised foam formulations for generation of antimicrobial PU foams [11]. The nature of this type of iterative DoE approach additionally reduces the need for tacit knowledge from a PU expert to guide formulation. Figure 1 shows the process flow for a DoE approach to formulation of materials adapted from Montgomery, 2017 [12].

The planning of experiments stage is particularly important. The first step is to define the problem and this requires that the objectives be fully developed [12].

This problem definition approach is essential when trying to understand a new formulation space or generate foams with wide ranging properties to optimise them for a novel application. One such application is the use of polyurethane foam as a synthetic growing media for crop growth in hydroponics [13,14,15]. Although PU foams have long been thought of as a suitable alternative to industrial growing media such as rockwool, with foam matching rockwool in terms of crop yield [13,14] and even increasing the yield of tomatoes grown using the nutrient film hydroponic technique [15] there is little literature on the optimum foam properties for maximising hydroponic crop yield. Previous work by the authors has shown that the addition of sodium bentonite to PU foams increased tomato vegetative growth in recirculated drip irrigation hydroponics and suggested that it was due to increased water holding capacity and the effect of the clay on cell morphology [16]. Furthermore, the authors showed that water holding content, foam cell size, and the ratio of open cells all affected the growth of spring onions in recirculated drip irrigation hydroponics [17]. There has, however, been no work linking the effect of cell morphology (cell size and open cell ratio) to the hydrodynamic (water holding, hydrophilicity and capillarity) of PU foams for this application.

The aims of this work are as follows: firstly, generate a set of PU foams with a large range of physical properties, particularly with regard to cell morphology and hydrodynamic behaviour, for application as a synthetic growing media using DoE principles; secondly, using these results to generate a semi-structured framework for a DoE approach to formulation of PU foams; and finally, use the responses from the DoE experiment to gain insight into relationships between PU foams’ cell morphology and their hydrodynamic behaviour.

## 2. Materials and Methods

### 2.1. Materials

Polyols, Voranol™ 1447 (a high ethylene oxide content polyether triol with a molecular mass of 4610) and Voranol™ 3322 (a high propylene oxide content polyether triol with a molecular mass of 3500) as well as the isocyanate, SpecFlex™ NE 112, a low functionality methylene diphenyl diisocyanate were kindly supplied by DOW Chemical Company (Midland, MI, USA). The surfactant Vorasurf™ 5906, a medium to high efficiency silicone siloxane, was also kindly supplied by DOW Chemical Company (Midland, MI, USA). The remaining surfactant Tegotab^®^ 8476, a silicone surfactant with excellent foam stabilisation with application in rigid foams, as well as the catalyst Dabco^®^ T (N-Methyl-N-(N,N-dimethylaminoethyl)-aminoethanol), a non-emissive amine catalyst that promotes the urea reaction, were kindly provided by Evonik Industries (Essen, Germany). The additive, Cloisite^®^ NE 116, a sodium bentonite clay, was kindly provided by BYK-Chemie GmbH (Wesel, Germany). Deionised water was used as the blowing agent and all reagents were used as received.

### 2.2. Methods

#### 2.2.1. Experimental Design

The primary aim of this work is to generate a set of PU foams with a large range of physical properties, particularly cell morphology and hydrodynamic behaviour, by following the experimental design process as shown in Figure 1.

##### Problem Statement

There is a lack of understanding about how to formulate a PU foam for use as a hydroponic growing medium, specifically how to formulate and model PU foam to generate a diverse range of cell morphologies and hydrodynamic properties.

##### Selection of Response Variables, Factors, and Factor Ranges

Eight main response variables are selected to help answer the problem statement. The first of these responses is chemical, the extent of reaction, see Figure 1, defined by the isocyanate conversion, which is a key response due to the application of the PU foam as a growing medium for crop growth. It is important to ensure that reagents (polyols, surfactants, catalysts) are all fully reacted into the polymer matrix to reduce the likelihood of any phytotoxic effects to crops, which have been noted in previous work and were attributed to mobile amine catalysts in formulations [18]. For this reason, a reactive amine catalyst was selected as the sole catalyst in our formulation. Two further kinetic responses were measured, foam height and “sigh back”/sag, and these relate to resulting foam properties. The cell morphology responses are cell size and the ratio of open cells. The hydrodynamic responses are water holding content, water drop penetration time, and two capillarity responses. As density can influence the economic viability of PU foam as a growing medium (growing media is sold on a volume basis, so lower density foams are potentially more profitable), it was also monitored as a response, however was not treated as a key response.

Although all the components in a polyurethane formulation are likely to influence the responses above, reducing the number of experimental factors drastically reduces the number of experiments required to explore the experimental space and allows for more detailed modelling of responses with an equal number of formulations. To minimise the number of experiments, only the amine catalysts and surfactants were selected as factors. PU foam catalysts play an important role in formulating polyurethane foams. It has been shown that adjusting catalyst compositions and loading changes the relative rates of the blowing and gelling reactions, which in turn significantly impacts PU foam properties [19,20]. The first factor selected, therefore, was a single reactive amine-based catalyst, Dabco^®^ T. Another key component of PU formulations are silicone surfactants. Silicone surfactants have a large effect on bubble generation and cell wall stabilisation of PU foams, and this results in a great impact on cell size and air permeability (or open cell fraction) of these foams [21]. Increasing surfactant loading has been shown to reduce cell size and the open cell content [22] and changing the composition of the silicone surfactant also influences cell structure, with an increase in surfactant backbone length being shown to reduce airflow (open cell fraction) [23]. Two surfactants with different chemical compositions and applications were chosen as factors: Vorasurf™ 5906 (used to make open-cell foams) and Tegotab^®^ 8476 (used to make insulating foam with mostly closed cells). Initial levels for these factors were selected from their data sheets and the PU formulation ranges are shown in Table 2. Isocyanate loadings varied slightly as an effect of balancing the stoichiometry, with isocyanate being supplied at a slight excess (isocyanate index = 1.15). The slight changes in isocyanate content were not expected to have any major effect on responses. The loading of the remaining components is kept constant. The initial formulation comes from earlier work [17] however many flexible PU foam formulations exist in literature and could be used as formulation starting points [24,25]. These remaining components are the polyisocyanate, polyols and water that make up the polymer backbone and this DoE studies the effects of the foam structure and morphology at the plant scale rather than the polymer chemistry at the molecular scale.

##### Choice of Experimental Design

A set of four experiments were designed to optimally answer the problem statement.

Experiment-1 was designed as a screen experiment, to determine the minimum amount of catalyst required for sufficient isocyanate conversion. Catalyst loading was the only factor and it was varied between 0 and 2.1. The only responses monitored was the kinetic responses, meaning these experiments could be completed in roughly 10 min per formulation. Eight loadings of Dabco^®^ T (Catalyst) were screened. The minimum loading required was then passed onto the next experiment. Table 3 shows the four experiments, their factors, responses as well as the time required per formulation. The range of catalyst loading to be used in the DoE experiment (experiment-4) was also determined in experiment-1.

Experiment-2 was designed to determine the minimum amount of Vorasurf™ 5906 (surfactant-1) required to make a stable foam and to determine the effect of surfactant-1 loading on kinetic responses as well as cell size. Surfactant-1 loading was the only factor in experiment-2. Surfactant-1 loadings were not equally spaced in experiment-2, with more points at low loadings and less experimental points at high loading, as at high loadings, surfactants can have little further effect on foam properties (particularly cell size) [26,27]. The kinetic and cell size responses are quick to measure, and each formulation took roughly 20 min. Seven loadings of surfactant-1 were screened and the minimum amount of surfactant-1 required to produce a stable foam was passed onto the next experiment. An appropriate range for surfactant-1 loading to be used in the DoE experiment (experiment-4) was also determined.

Experiment-3 introduced surfactant-2 as the only factor to determine the effect of this surfactant on the kinetic responses as well as the two cell morphology factors (cell size and open cell ratio). Similar to experiment-2, Surfactant-2 loadings were not equally spaced in experiment-3 with more points at low loadings and less experimental points at high loading. Catalyst and surfactant-1 loadings were held constant and determined in Experiment-1 and 2. The addition of the open cell ratio test meant that each screen took roughly 30 min and eight formulations were tested. An appropriate range for surfactant-2 loading to be used in the DoE experiment (experiment-4) was also determined.

In experiment-1, 2 and 3 a subset of responses were measured to save on experimental time and to quickly screen for appropriate catalyst and surfactant ranges in experiment-4. Figure 2 shows the iterative nature of these four experiments and the information that was passed on to subsequent experiments. A further advantage of this sequential approach is that it greatly reduced the likelihood of a formulation in the DoE experiment producing a foam that catastrophically fails due to insufficient catalyst or surfactant.

Experiment-4 used a DoE approach and all responses were analysed using a second order response model of the form shown in Equation (1),
(1)$$y=\underset{1\leq i\leq q}{\sum }\beta_{i}x_{i}+\underset{1\leq i\leq j\leq q}{\sum }\beta_{\mathrm{ij}}x_{i}x_{j}+\underset{1\leq i\leq q}{\sum }\beta_{\mathrm{ii}}{x_{i}}^{2}+\varepsilon$$
where q is the number of factors (x), y is the response, β_i_ is the fitting coefficient and ε is the random error parameter. Whilst ε is known as the random error parameter, it also accounts for the contribution of all the other components in the PU foams which are kept constant. x_i_x_j_ is an interaction parameter, accounting for any synergism or antagonism between factors and x_i_^2^ is a curvature parameter. The number of parameters in Equation (1) when there are three factors is 10, the error parameter, the three main effects, three interaction effects and three curvature effects. The minimum number of experiments required to fit an empirical model needs to be greater than the number of terms in the fitting model, which means we need a minimum of 11 formulations for a second order response model. For experiment-4 it was decided to do 16 formulations to account for experimental errors and in case any of the foams catastrophically failed. JMP^®^ Pro 16 was used to determine the optimum set of formulations and a D-optimal design (D-efficiency = 40.52, G-efficiency = 64.26 and A-efficiency = 21.99) was used to generate the formulations. Formulations for all foams (experiment-1, 2, 3 and 4) are available in Appendix A.

##### Statistical Analysis of the Data

Select responses were measured for each experiment and when responses were measured in replicates, the mean value was used for modelling. The generalised model was used to fit data, however reducing the complexity of models is an important step in data fitting. This reduction in model complexity is important as several factors may not influence the desired response and are therefore not required in the model. This reduces the risk of overfitting and simpler models are more likely to accurately represent the process. One method for doing this is the use of stepwise linear regression. This method fits each possible variation of Equation (1) by adding or removing one variable at a time using the variable’s statistical significance [12]. K-fold cross validation is a technique that maximises the robustness of models in small data sets. The technique randomly divides data into k subsets, using each subset once as a validation set and all remaining data as the training set to fit the model [28]. The model giving the best validation statistic (average r^2^ for all subsets) is then selected. Using linear regression whilst maximising the k-fold leads to robust models, with low likelihood of overfitting. A k-fold subset value of 5 was used. All residuals were checked for normality and homogeneity of variance.

#### 2.2.2. PU Foam Generation

The polyisocyanate was accurately weighed into a 30 mL syringe. The remaining reaction components were weighed into a 568 mL polypropylene cup and mixed at 3000 RPM for 45 s with an overhead mixer with a straight blade disk agitator. This mixture was allowed to debubble in a fume hood for 5 min before reacting. The polyisocyanate was added to the polypropylene cup and further and mixed at 1500 RPM for 6 s using the same overhead mixer/stirrer combination. The reacting mixture was immediately transferred to the reaction vessel in the FoamPi for 10 min before being transferred to a curing oven at 120 °C for 20 min.

#### 2.2.3. Isocyanate Kinetics

PU Foam reactions took place in the FoamPi [29] reaction vessel, an apparatus developed by the authors for measuring foam reaction kinetics. The FoamPi logs temperature rise, height change and mass change during the reaction. Due to the nature of the exothermic reactions and with the use of an adiabatic temperature rise correction, isocyanate conversion can be determined from the increase in temperature [30]. Details of this calculation are given in the FoamPi paper [29].

Maximum foam height measured using the FoamPi and the normalised maximum height (height divided by reagent mass) is reported here.

Foam sag or “sigh back” is also calculated using data from the FoamPi and is reported as the percentage reduction in height from the maximum height to the final height as shown in Equation (2).
(2)$$\mathrm{Sag}=100\;\times \;(1-\frac{h_{\mathrm{final}}}{h_{\max}})$$

#### 2.2.4. Foam Physical Properties

Foam density was measured according to ASTM D3574-11 test A [31], a piece of foam measuring 25 × 50 × 50 mm was cut, accurately measured with a digital Vernier calliper and the mass recorded. Density was calculated from the foam mass and volume.

Cell size was calculated using an adaptation of the method ASTM D3576-15 [32], a piece of foam was cut perpendicular to the rise direction and the surface was stained using a marker pen, then imaged using optical microscopy. ImageJ [33] software (Bethesda, MD, USA) was used to determine individual cell size, for each sample a minimum of 200 cells are counted. The mean cell diameter and standard deviation are reported.

Airflow through the foam was calculated using ASTM D3574-11 test G [31], whereby the air flowrate at a constant pressure (125 Pa) is recorded in l min^−1^. This airflow measurement can be used as a measure to determine the effective open cell content of PU foams [34].

Water holding content (WHC) is measured by submerging a 25 × 50 × 50 mm of foam in water for 24 h. Samples are then removed from the water and allowed to drain freely for 15 min before measuring their mass. The WHC is calculated from the difference in sample wet and dry mass, which is divided by the volume and is reported as g_water_ dm_foam_^−3^.

Water drop penetration test (WDPT) is a simple and useful test in soil science [35] to determine water uptake in soils and was measured by dropping a drop of stained (1% bromophenol blue) deionized water onto the foam surface from a height of 1 cm and determining the time taken for this drop to be fully absorbed. This was repeated 5 times and the mean WDPT is reported. Foams could be further classified into hydrophilic rankings using this WDPT [36].

Capillarity of the foams was measured using an adaptation of the apparatus described by Schulker et al. [37]. Briefly, foam samples were cut into 20 mm × 20 mm × 50 mm pieces and placed vertically in a sub irrigating system which irrigated the foam samples for known time periods to a height of 25 mm. This irrigation is repeated over several cycles to generate a water uptake curve. Sample mass was determined between each irrigation cycle. The height of water absorbed could be calculated from this mass difference. Triplicate samples were measured and the mean capillary water rise height for each time period is reported. Furthermore a water uptake curve is generated from this data and an exponential decay curve is fitted to this. The fitting parameters of this curve give important insight into the capillarity of the foam samples. Equation (3) shows the equation used to fit the capillary rise data.
(3)$${\mathrm{CRH}=\alpha }_{1}(1-e^{\alpha_{2}t})$$
where CRH is the capillary rise height in cm, α_1_ is the maximum capillary rise height in cm, α_2_ is the rate of water uptake in cm s^−1^ and t is the total sub irrigation time in seconds.

#### 2.2.5. Statistics

Determination of the DoE formulation points was done using JMP^®^ Pro 16 SAS Institute Inc., Cary, NC, USA, 1989–2022. Modelling of physical property responses in experiment-4 was also done in JMP^®^ Pro 16 using stepwise linear regression and k-fold cross validation (k = 5) with a hereditary restriction.

Model fitting for experiments 1, 2 and 3 was done using the numpy (polyfit) [38] and scipy (optimize.curve_fit) [39] packages in Python 3.8. The scipy (optimize.curve_fit) package was also used for fitting capillarity data in experiment-4.

## 3. Results and Discussion

### 3.1. Experiment-1: Catalyst (Dabco^®^ T) Screen

Eight formulations were tested in experiment-1, with catalyst loading being the only factor and with kinetic responses as the only recorded responses. Isocyanate conversions, normalised foam height, and time to maximum foam height were the kinetic data measured. The full formulation data for experiment-1 is available in Appendix A.

Temperature rise data is used to calculate isocyanate (-NCO) conversion, and the relationship between isocyanate conversion and catalyst loading is shown in Figure 3A. The conversion increases rapidly from ~0.55 with zero catalyst to a conversion of ~0.95 at a catalyst loading of 0.5 PPHP. Any further increase in catalyst does not increase the conversion. Formulations with catalyst loadings above 1.5 PPHP led to rapid reactions where the mixture was difficult to transfer to the reaction vessel. These are excluded from Figure 3A. An exponential curve is fitted to the data, and this fit has a horizontal asymptote at X_NCO_ = 0.977, which indicates the maximum possible -NCO conversion for this system.

The normalised maximum height increased linearly with an increase in catalyst loading as shown in Figure 3B, and the time to maximum height decreased exponentially as shown in Figure 3C. The asymptote of the time to maximum height curve is at 50.9 s, the minimum time to reach maximum height.

The conversion data confirms that the single amine-based catalyst (Dabco^®^ T) is sufficient to achieve complete isocyanate conversion (X_NCO_ > 0.90) in these formulations at all loadings above 0.5 PPHP. The linear increase in normalised maximum height indicates that the catalyst is likely to influence physical foam properties, particularly foam density. To explore this effect, catalyst loading will to be screened in experiment-4. The selected factor range for the catalyst used in experiment-4 is between 0.5 PPHP (the minimum required for 90% -NCO group conversion) and 1 PPHP (the maximum catalyst loading before mixing and transferring to the reaction vessel becomes difficult due to rapid reaction times). For experiments 2 and 3, where the catalyst loading remains constant, an intermediate loading of 0.8 PPHP was used. This value was the amount required for 95% NCO group conversion. These eight short screening formulations (10 min experimental time per formulation) give significant insight into the catalytic effect of the single catalyst, confirming it is suitable as the sole catalyst and guiding the selection of catalyst loading for successive experiments.

### 3.2. Experiment-2: Surfactant-1 (Vorasurf™ 5906) Screen

Seven formulations were screened in experiment-2 with surfactant-1 loading being the only factor. The response variables measured were the same kinetic responses as in experiment-1 as well as the percentage sag or “sigh back”, calculated using height data, an indicator of cell opening of a PU foam [40]. Cell size, one of the targeted cell morphology responses, is also measured. The full formulation data for experiment-2 is available in Appendix A.

Figure 4A shows the -NCO group conversion with the dashed line indicating the predicted -NCO conversion at a catalyst loading of 0.8 PPHP (X_NCO_ = 0.95). At a loading of 0 PPHP surfactant-1 the foam collapsed and never reached the temperature probe, hence the low value of X_NCO_. At loadings between 0.25 PPHP and 0.75 PPHP, conversion is lower than expected, ranging between 0.87 and 0.91. Above 0.75 PPHP, X_NCO_ is within 0.03 of the expected conversion, which is within the precision of the FoamPi temperature probe and therefore not significant.

Figure 4B shows the normalised maximum height data, which exhibits exponential behaviour with regard to surfactant-1 loading, and the fitted model predicts a maximum normalised height of 3.40 mm g^−1^ using this surfactant in this specific formulation. At loadings above 1 PPHP surfactant-1, maximum normalised height does not increase.

Percentage sag or “sigh back” data is shown in Figure 4C. The percentage sag decreases from 30% at 0 PPHP surfactant-1 to 7.3% at 0.75 PPHP surfactant-1. This decrease indicates the stabilising effect of the surfactant, reducing the “collapse” of the foams whilst the remaining foam is left with an open cell morphology (as there is still some sag). Above 0.75 PPHP, the sag increases slightly to a value of 12% at a loading of 4 PPHP surfactant-1, which is unexpected and could indicate shrinkage due to closed cells. However, this would need to be confirmed by microscopy/airflow measurement.

Figure 5A shows the effect of surfactant-1 concentration on the cell size of the foam with the error bars indicating one standard error of the cell size as calculated from more than 200 cells. Figure 5B–G show optical images of the stained foam surfaces at increasing surfactant-1 loading. The cell size decreases, from a cell size of 1200 ± 76 µm at 0.25 PPHP loading surfactant-1, until it plateaus at a loading of 1 PPHP surfactant. An exponential fit of the data shows that the predicted minimum cell size with this surfactant in this formulation is 678 µm. The error bars also reduce in size, indicating that cell size distribution becomes more uniform with an increase in surfactant-1 loading. The images on the right show optical microscopy images of the surface marked foam.

From the results of experiment-2, an understanding of the kinetic effects as well as some information on cell morphology have been gathered for sequential experiments. The minimum amount of surfactant-1 required to produce a stable foam is 0.5 PPHP. As we are interested in reducing the cell size of PU foam to increase capillary action, the range of interest is that which produces the smallest cell sizes. For experiment-3 a loading of 0.8 PPHP is selected, where the predicted cell size is expected to be 753 µm. The range selected for experiment-4 is between 0.8 PPHP (stable foam with reduced cell size) and 2 PPHP (highly stabilised foam with small and uniform cell size). These seven relatively quick screening formulations (20 min experimental time per formulation) give particularly useful information into the effect of surfactant-1 on PU foam cell size.

### 3.3. Experiment-3: Surfactant-2 (Tegostab^®^ 8476) Screen

Experiment-3 screened surfactant-2, Tegostab^®^ 8476, a silicone surfactant that promotes extremely small and closed cell morphology. This was the only factor in this screen. The response variables were the same as those in experiment-2 with the addition of open cell content, calculated using airflow data, the second targeted cell morphology response. The full formulation data for experiment-3 is available in Appendix A.

Figure 6A shows X_NCO_, the isocyanate conversion at varying surfactant-2 loadings. The conversion is reduced at all loadings of surfactant-2, X_NCO_ varied between 0.872 and 0.929, and although this reduction is slight, it may indicate an antagonistic effect between surfactant-2 and the catalyst, which can be further explored in the DoE experiment-4.

Figure 6B shows the normalised maximum height, which increases with any addition of surfactant-2 above the baseline of 3.47 mm g^−1^. This reaches a plateau at a loading of ~1 PPHP and a normalised maximum height of 3.99 mm g^−1^ is predicted to be the greatest value of this formulation using an exponential fit of the data. This increase in normalised height shows the greater stabilising ability of surfactant-2 when compared to surfactant-1.

Sag is shown in Figure 6C, which reduces with increased surfactant-2 loading. Interestingly, at low levels of surfactant-2 sag is greater than the formulation with none of this surfactant. This may indicate that at low loadings of surfactant-2 there is an antagonistic effect due to incompatibility between the two surfactants. At all loadings above 1 PPHP surfactant-2 the sag is reduced below that of the formulation with no surfactant-2. An exponential fit of the data predicts a minimum sag of 2.90%.

The effect of surfactant-2 loading on cell size of the foam is shown in Figure 7A with error bars indicating one standard error with more than 200 cells counted for each sample. Figure 7B–I shows optical images of the stained PU foam surface at increasing surfactant-2 loading. At 0.1 PPHP surfactant-2 we see a reduction in cell size from the formulation with no surfactant-2. However, at loadings between 0.2 PPHP and 1 PPHP cell size is not significantly different to the foam without this surfactant. At loadings of 2 PPHP, 3 PPHP and 4 PPHP the cell size reduces to 610 ± 13.4 µm, 549 ± 9.69 µm and 558 ± 8.10 µm, respectively. These results indicate that the cell size does not reduce between 3 PPHP and 4 PPHP and that these values likely represent the smallest cell size that can be produced using this surfactant in this foam formulation. The low standard error values indicate that foams with highly homogenous cells are produced using this surfactant, particularly at higher loadings.

Airflow data is shown in Figure 8A. Airflow decreases with any increase of surfactant-2 and reaches a plateau at surfactant-2 loadings above 2 PPHP. From previous work (data not shown), airflow values above 130 L min^−1^ indicate a foam with an open cell ratio exceeding 0.95. All formulations with a loading of surfactant-2 below 0.5 PPHP fall into this category. Figure 8B shows the effective open cell ratio (p_eff_) calculated from the airflow data for formulations with surfactant-2 loadings above 0.5 PPHP. In the range between 0.5 PPHP and 4 PPHP surfactant-2 we can produce a range of foams where p_eff_ ranges between 0.13 and 0.85. The exponential fit of this data predicts a minimum open cell ratio of 0.12.

In experiment-3 eight screening formulations with varying loadings of surfactant-2 were trialled. These eight formulations required roughly 30 min each to react and measure the resulting foam morphology properties. This set of formulations produced a wide range of PU foam cell morphologies. There was little effect on the cell size of the foams at surfactant-2 loadings below 0.5 PPHP and there was possibly some antagonism between the two surfactants at low loadings as sag increased from the formulation with no surfactant-2. Surfactant-2 reduced airflow and increased the closed cell content as expected from this surfactant. Considering the range of catalyst (0.5–1 PPHP) and the range of surfactant-1 (0.8 PPHP–2 PPHP) suggested for experiment-4, and the catalyst and surfactant loadings used in experiment-3 (0.8 PPHP and 1 PPHP, respectively), the range of surfactant-2 needs to be carefully selected for experiment-4. The lower bound is selected as 1 PPHP surfactant-2. To account for the case where high blowing catalyst and surfactant-1 loading would tend to a high open cell content, this value would lead to a foam with mostly open cells. The upper range was selected to be 3 PPHP as any further increase in surfactant-2 had no effect on cell size and resulted in a mostly closed cell foam.

### 3.4. Experiment-4: DoE Screen

Experiment-4 focused on mapping the experimental formulation space defined by the three screening experiments that preceded it and fully characterising the resulting PU foam physical properties. The DoE factors are the catalyst, Dabco^®^ T (range 0.5–1 PPHP), surfactant-1, Vorasurf™ 5906 (range 0.8–2 PPHP), and surfactant-2, Tegostab^®^ 8476 (range 1–3 PPHP). The responses are isocyanate conversion (X_NCO_), density (ρ), cell size (d_cell_), open cell fraction (p_eff_), water holding capacity (WHC), water drop penetration time (WDPT), maximum capillary rise (α_1_), and rate of water uptake (α_2_). 16 formulations were screened. For modelling the three factors are labelled as x_1_ (catalyst), x_2_ (surfactant-1) and x_3_ (surfactant-2). The full formulation data for experiment-4 is available in Appendix A.

The three screening experiments led to all 16 formulations producing stable foams, which would have been highly unlikely if an iterative DoE approach had not been taken.

Figure 9 shows model parameters for each of the kinetic and cell morphology responses. The top part of each subplot shows the significant model parameters from the generalised model used to predict each of the responses. The horizontal bars indicate the value and sign of the result of the t-ratio for each model parameter. The t-ratio is defined as the estimate of the coefficient divided by the standard error of the estimate. Those shown in pink have a t-ratio > 2 which indicates the parameter has *p* < 0.05. The bars in blue indicate that the value of the t-ratio < 2 and *p* > 0.05. The plot below the horizontal bars shows the actual response as a function of the predicted response, with the fit shown in the solid pink line and the dashed black line indicating the y = x line. The r^2^ is inset as is the k-fold r^2^ (k = 5) for each of the models.

The t-ratio and actual/predicted isocyanate conversion (X_NCO_) are shown in Figure 9A. The random error parameter, ε, is the parameter that explains the most variance, indicating that at the lower limit of factor ranges (x_1_ = 0.5 PPHP, x_2_ = 0.8 PPHP, and x_3_ = 1 PPHP), there is an isocyanate conversion of 0.849. This is lower than the expected conversion at a catalyst loading of 0.5 PPHP (0.900) and may be explained by the value of the surfactant-2 coefficient (x_3_) as well as the negative value of the interaction parameter between x_1_ and x_3_. These show that x_3_ reduces isocyanate conversion and that catalyst loadings may need to be increased slightly in the presence of x_3_. This confirms the X_NCO_ results from experiment-3, where conversion was lower in all formulations than expected. Furthermore, the X_NCO_ model, whilst adequate, underpredicts at low conversions, as can be seen from the deviation from the x = y curve in Figure 9A. The model has four fitting parameters as well as the random error parameter and is only influenced by the loading of the catalyst and surfactant-2. The r^2^ value of the model, 0.746, is low, and the model is not very robust with a low k-fold r^2^ of 0.403. These low values can be explained by the fact that all formulations in experiments 2, 3, and 4 were at values of catalyst loading that should have led to X_NCO_ values above 0.9. At these values, the response is more sensitive to any experimental errors, either in weighing of components or errors introduced by the FoamPi apparatus used in measuring temperature rise. The FoamPi thermocouple had a relative standard deviation of 3.33% in terms of maximum temperature, and this may explain the variation, which is unexplained by the model.

The t-ratio and actual/predicted density (ρ) are shown in Figure 9B. The random error parameters, ε, explains the greatest amount of variation in the density model. The density only varies between 25.4 kg m^−3^ and 29.6 kg m^−3^, and this low variation, much like the X_NCO_ result, increases the significance of any experimental error. The model has four fitting parameters as well as the random error parameter and is only influenced by the loading of the two surfactants. The density model has the lowest r^2^ value of all modelled responses (0.699) and a low k-fold r^2^ of 0.481. x_3_, surfactant-2 loading, explains the most variation after the random error parameter and is negative, reducing the foam density with increased loading. x_2_ is also negative, showing that increasing the loading of either surfactant reduces density (however, this falls below a t-ratio of 2, indicating *p* > 0.05 of this coefficient), and the interaction parameter indicates that there is a synergism for decreasing density between the two surfactants. The model underpredicts at low density and overpredicts at high density, observable in the deviation from the x = y curve in Figure 9B. Although some insights can be taken from this model, it shows the importance of formulating for the generation of foams with a large range of targeted physical properties.

The t-ratio and actual/predicted cell size (d_cell_) are shown in Figure 9C. Cell size is the first of the targeted physical properties. The random error parameter, ε, again explains the highest variance. At the minimum loadings of the three factors, the model predicts a cell size of 602 µm. Cell size is predicted using all three factors as well as two interaction parameters, between the catalyst loading and the surfactant loadings. The model has five parameters as well as the random error parameter. The cell size is reduced by increasing the loading of catalyst as well as surfactant-2, which is as expected. Surfactant-1 has a positive coefficient, indicating that an increase in loading increases cell size. However, the low t-ratio value (1.55) means this coefficient has a *p* > 0.05. The low significance of this coefficient may be due to the range used for this surfactant-1. In experiment-2, we determined that cell size did not vary drastically above loadings of 1 PPHP. The range of 0.8 PPHP and 2 PPHP in experiment-4 means that surfactant-1 loading had little effect in the ranges used in this experiment. The high negative t-ratio values of the interaction coefficients between catalyst loading and both surfactants indicate that increasing catalyst loading, whilst increasing the loading of either surfactant, acts synergistically and decreases the cell size. The r^2^ value of 0.868 shows a good fit of the model, and the k-fold r^2^ = 0.664 shows that the model is reasonably robust. Finally, the model fits well to the x = y curve shown in Figure 9C, indicating that the model does not over or under predict cell size at any of the values in this experiment.

The t-ratio and actual/predicted open cell fraction (p_eff_) are shown in Figure 9D. The random error, ε, explains most of the variance, and the model predicts that at the minimum range of loadings for the three factors, there is an open cell fraction of 0.655. Open cell fraction is the second of the targeted cell morphology properties and the response is a function of all three of the factors as well as the curvature parameter in x_3_. The model has a total of four fitting parameters as well as the random error parameter. The surfactant-2 explains the second highest amount of variance and has a negative coefficient, reducing the open cell fraction of the foams. This is expected as this surfactant is an excellent cell stabiliser used in the production of rigid foams. Surfactant-1 is also an important parameter in the model, with a positive coefficient, indicating that an increase in surfactant-1 increases the open cell content. Again, this is not unexpected as it is a surfactant that promotes open cells. The curvature parameter in surfactant-2 means that as you increase the loading of surfactant-2, the rate of reduction of the number of open cells decreases. This is also expected since there is a hard limit to the fraction of closed cells. An open cell ratio below zero or above one cannot exist in a foam. The negative value of the catalyst coefficient is unexpected as this catalyst promotes the blowing reaction and it would be expected that increasing the rate of blowing reaction would increase the number of open cells. The r^2^ of the model is good (0.931) and the k-fold r^2^ = 0.900 indicates that this model is highly robust in this dataset. The model also predicts values well within the range of responses, as shown by the overlap between the model fit and the x = y line in Figure 9D.

Like Figure 9, Figure 10 shows model parameters for each of the hydrodynamic property responses.

The t-ratio and actual/predicted water holding content (WHC) are shown in Figure 10A. WHC is the first of the targeted hydrodynamic properties. The random error parameter again explains the greatest amount of variance. This translates to a WHC of 1030 g dm^−3^. This value is above the theoretical maximum (1000 g dm^−3^) and indicates that the foams are likely swelling to a volume above their dry volume (the polymer itself swells in water and the foam struts expand). Therefore, this response WHC is likely acting as a combination response, combining the maximum water holding as well as the maximum swelling of the foam, which may be of interest. Therefore, it is reported as is (g water per dm^−3^ dry foam), allowing for values above the threshold of 1000 g dm^−3^. The model consists of four fitting parameters as well as a random error parameter. The WHC ranges between 820 g dm^−3^ and 1130 g dm^−3^, indicating that over 24 h, all formulations, even those with a high number of closed cells, can absorb a large amount of water. The factor explaining the highest amount of variance is surfactant-1 loading, which has a positive coefficient indicating that an increase in surfactant-1 increases WHC. The surfactant-2 coefficient is negative, as is that for the catalyst. This result is similar to the model for open cell fraction and may indicate that open cell fraction may be a good predictor for WHC. The r^2^ for the model is very good (0.927) and the k-fold r^2^ = 0.801 shows that this model is robust in this dataset. The model also predicts values well within the range of responses, as shown by the overlap between the model fit and the x = y line in Figure 10A.

The t-ratio and actual/predicted water drop penetration test (WDPT) are shown in Figure 10B. The WDPT is the second target response for the hydrodynamic properties. It is only a factor of the two surfactant loadings, with the loading of surfactant-2 explaining the most variance. The positive value of this coefficient indicates that increasing surfactant-2 loading increases the WDPT. This is not unexpected as this surfactant increases the number of closed cells, which should increase the time taken for water to penetrate the foam. Surfactant-1 loading has the second highest significance and a positive value, as expected for this surfactant that promotes open cells. The interaction parameter between the two surfactants shows that at high loadings of surfactant-1, surfactant-2 has a lesser effect on the WDPT. The last two factors, the curvature of the surfactant-2 coefficient and the random error parameter, have a t-ratio of less than 2 and therefore a *p* > 0.05. This model has the worst fit of the targeted responses, with an r^2^ of 0.791, and is the least robust, with a k-fold r^2^ of 0.518. This poor fit may be due to the nature of the WDPT, which has a large variation between samples due to the subjectivity of determining the endpoint of the experiment, particularly with formulations with longer WDPT times. For example, one formulation had a mean WDPT of 61.5 s but varied between 14.9 s and 126.7 s in the five repeat droplets. For future experiments, it may be required to increase the number of repeat droplets to reduce this error or use a more robust test. The model also overpredicts the WDPT at high WDPT times, as shown in the deviation from the x = y line in Figure 10B.

Figure 10C,D show the t-ratio and actual/predicted values for α_1_ and α_2_ coefficients, the two fitting parameters for the capillarity test. These two parameters are expected to be the most important in predicting plant growth in hydroponic experiments and are the responses of most interest here. The maximum capillary rise required the most complex model to fit the data. This model has six fitting parameters as well as a random error parameter and is influenced by all three of the factors. The random error parameter explains the largest amount of variance and predicts a maximum capillary rise of 3.60 cm at the minimum loadings of the three factors. The two surfactants had the same effects as those seen in the previous hydrodynamic tests, with an increase in surfactant-1 increasing the maximum capillary rise and an increase in surfactant-2 reducing the maximum capillary rise. This again suggests that open cell fraction may influence the maximum capillary rise. The effect of catalyst loading had a t-ratio below 2 and therefore a *p* > 0.05, so it had little effect on the maximum capillary rise. The interaction coefficient between all three factors was significant, indicating that at high catalyst loadings, the two surfactants have a larger effect on maximum capillary rise, while surfactant-2 loadings have a smaller effect at high surfactant-1 loadings. The fit for maximum capillary rise is good with a r^2^ of 0.918 and is reasonably robust with a k-fold r^2^ of 0.667. The fit predicts values of the maximum capillary rise with the fit having little deviation from the y = x curve Figure 10C.

The greater the absolute value of the rate of water uptake, the greater the curvature in the exponential fit of the water uptake curve and the quicker the time until the maximum capillary rise is reached. The model for the rate of water uptake (α_2_) required four fitting parameters as well as a random error and was influenced by all three of the factors. The random error parameter explained the largest part of the variance. The effects of the two surfactants are the same as for the other hydrodynamic properties, with an increase in loading of surfactant-1 increasing the rate of water uptake and an increase in loading of surfactant-2 reducing the rate of water uptake. An increase in catalyst loading decreased the rate of water uptake. The signs of these three coefficients in the model are the same as those that predicted open cell content, suggesting that open cell content may have an important role in predicting the rate of water uptake. The model explains a large fraction of the variance and has an r^2^ = 0.920 and a k-fold r^2^ = 0.804 suggesting that the model is robust within this dataset. Finally, the model does not over or under predict the rate of water uptake as there is little deviation from the x = y curve and the predicted/actual fit shown in Figure 10D.

### 3.5. Effect of Cell Morphology on Hydrodynamic Properties

The results from the modelling of the physical properties showed that the sign and t-ratio of the coefficients in cell morphology models, particularly open cell fraction, and hydrodynamic properties models were often similar. It is therefore worth examining these relationships further.

A generalised model of the form shown in Equation (4) was used to fit the cell morphology parameters (cell size and open cell fraction) to the hydrodynamic properties.
(4)$${\mathrm{HP}=\beta }_{1}p_{\mathrm{eff}}{+\beta }_{2}d_{\mathrm{cell}}{+\beta }_{3}{\;p}_{\mathrm{eff}}^{2}{+\beta }_{4}{\;d}_{\mathrm{cell}}^{2}{+\beta }_{5}p_{\mathrm{eff}}d_{\mathrm{cell}}+\varepsilon$$

HP is the hydrodynamic parameter of interest (WHC, WDPT, α_1_ or α_2_) and p_eff_ and d_cell_ are the cell morphology parameters. The generalised model was simplified by removing non-significant parameters using the same approach as that used in experiment-4 for modelling foam responses from catalyst and surfactant loadings. This model proved a good fit for the WHC and α_2_ properties, but none of the possible models fitted the WDPT and α_1_ responses well. On examination, a better fit for those response was an exponential fit of the form shown in Equation (5).
(5)$${\mathrm{WDPT}=\beta }_{1}e^{\beta_{2}p_{\mathrm{eff}}}+\varepsilon \;$$

For all models, all the coefficients for cell size (d_cell_) were not significant and were dropped from the models. The open cell ratio explained a large proportion of the variation in all four of the hydrodynamic properties. This may be due to the relatively low range of cell sizes in experiment-4 (~550 µm to 700 µm). Figure 11 shows the four hydrodynamic properties as a function of the open cell fraction.

The WHC is shown in Figure 11A with a quadratic equation fitted to the data. As the open cell content increases, the WHC also increases. However, at higher open cell content, this reaches a maximum and the WHC does not appreciably increase further. The r^2^ of the fit is 0.799. The WDPT as a function of p_eff_ is shown in Figure 11B with the exponential fit of the data. The exponential increase in WDPT at low open cell fraction is an important finding showing that there is a rapid increase in WDPT or “hydrophobicity” of the foam as we introduce closed cells that only take up water slowly following swelling of the polymer. The exponential fit had an r^2^ of 0.804. Practically, at these high WDPT (>60 s), they would be classed as “slightly hydrophobic” soils [36] and would likely not be suitable for use as synthetic growing media. We can therefore conclude from the model that at p_eff_ < 0.146 foams (at which WDPT = 60 s) are not suitable as synthetic growing media.

Figure 11C shows the maximum capillary rise (α_1_) data as a function of open cell ratio and shows the exponential fit to the data. This exponential fit with a r^2^ of 0.886 outperformed the best fit of the generalised model (Equation (4)) which had an r^2^ of 0.599. This model shows that increasing the p_eff_ value increases the maximum capillary rise to a maximum (3.40 cm) at an open cell fraction of 0.336 and increasing the open cell fraction further does not increase the maximum capillary rise further.

Figure 11D shows the rate of water uptake (α_2_) data as a function of open cell ratio and shows a linear fit of the open cell ratio to α_2_. This fit explains a significant amount of the variance with an r^2^ of 0.882. This finding indicates that any closed cells impede the pathway for capillary action, reducing the rate at which water can be taken up by the foam via capillary action. The significance of the results shown in Figure 11A,C,D is that there may be a trade-off between the rate of water uptake, maximum capillary rise and water holding content, with all three of these properties being highly influenced by the open cell fraction. It is not known which of these growing media properties is more important in regulating plant growth in hydroponic systems. However, with the models and formulations developed in this work, these experiments can now be done.

## 4. Conclusions

We have shown that with 23 screening formulations and 16 final formulations and by only varying catalyst and surfactant loadings, we are able to gain significant insight into the experimental space of a PU foam formulation using a semi-structured DoE framework. This was done in a set of sequential experiments. In experiment-1, we used the catalyst (Dabco^®^ T) loading as a factor and determined that this single catalyst was sufficient to produce stable foams in this formulation, and at catalyst loadings above 0.5 PPHP, we could predict an -NCO conversion above 90%. In experiment-2, we examined the effect of surfactant-1 (Vorasurf™ 5906) on kinetic responses as well as cell size. The surfactant had little effect on NCO conversion but increased the normalised foam height and decreased sag, or “sigh back”. Importantly, an increase in surfactant reduced cell size and at loadings above 0.8 PPHP the cell size did not decrease further. An exponential fit of the data predicted a minimum cell size of 680 µm using this surfactant in this formulation. In experiment-3, we introduced a second surfactant (Tegostab^®^ 8476), a surfactant with cell window stabilising ability used in the production of closed cell foams, and examined its effect on kinetic responses as well as cell morphology (cell size and open cell fraction). This surfactant reduced NCO conversion slightly, increased the normalised foam height, and reduced sag. It also reduced the cell size when used at a loading above 1 PPHP, but did not decrease it further when used above 3 PPHP. An increase in the loading of surfactant-2 reduced the open cell content, and above a loading of 0.5 PPHP, the open cell fraction dropped below 0.9. At loadings above 3 PPHP, the open cell fraction reached a minimum of 0.12. In experiment-4, a DoE approach was used to model eight responses, using three factors (catalyst loading, surfactant-1 loading and surfactant-2 loading) and factor ranges determined in experiments 1, 2 and 3. Sixteen formulations were tested, all of which produced stable foams with a large range of targeted physical properties. This approach proved powerful as all eight responses were successfully interpreted using a generalised model that was reduced to only include significant effects. The models of targeted responses (cell size, open cell content, WHC, maximum capillary rise, and rate of water uptake) explained the most variance and were the most robust. Finally, it was shown that the hydrodynamic properties could be modelled using only the open cell fraction, showing the importance of this cell morphology parameter in determining the water absorption parameter of PU foams.

This sequential semi-structured DoE approach to formulating PU foams has been shown to be successful in generating a huge amount of information on the formulation space using less time and resources, and with the use of several screening experiments, the amount of “tacit” formulation knowledge is reduced. Furthermore, this kind of DoE approach could easily be adapted depending on the formulation space and properties of interest.

## Data Availability

All primary and Appendix A used in this study is available under a CC BY 4.0 license and is available at https://doi.org/10.15131/shef.data.21510876 (accessed on 23 November 2022).

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
