# Peer review of "Rational Design of a Polyurethane Foam"

_polymers, 2022, doi:10.3390/polym14235111_

Round 1

Reviewer 1 Report

The manuscript entitled “Rational Design of a Polyurethane Foam" generates a semi-structured framework model and uses the Design of Experiment (DoE) method to formulate a PU foam with optimum foam properties for maximizing the hydrodynamic behavior.

1. Most of the citations are old. The introduction can be polished and cite some of the recent work of the PU foam research. 

2. the study of cell size of polyurethane, can the authors elucidate more on this part? e.g., why are there fewer experiment points selected between 2 to 4 PPHP compared with the data between 0-1 PPHP?  Figure 5 & 7 should have the index of the subfigures and it's better to explain the optical images in the figure title context.

3. Figure. 11 (C) for the alpha1 fitting, since the quadratic function doesn't look that good. Is there any better way for the fitting? if r^2 is just 0.599, how can the authors validate the derived relation is reasonable? Is it possible to use some data analysis method like KNN for the fitting?

Author Response

Dear Reviewer, 

Thank you for your time and effort in reviewing our manuscript. In response to your comments.

(A) Added experimental design of PU foam manuscripts that detail the use of a DoE approach in designing foam for different applications, with manuscripts from 2017-2022.

(B) It was expected that there may be a diminishing effects on PU properties as surfactant loading increased and major effects would be seen at lower loadings. Therefore it was decided to have smaller increments at lower loadings. This reasoning has been added to the experimental design section for experiment 2-3. Figures 5 and 7 have been changed to have indexes and these are referred to in text.

(C) We agree that the low r2 may indicate that there may be a more suitable fit. On further examination the use of an exponential model, in the same form as that used in 11(B) is a much better fit with an r2 of 0.886. The figure and description have been updated to show this. With this higher r2 we do not believe that more complex modelling, such as KNN regression is not necessary. 

Reviewer 2 Report

The paper deals with a methodology of polyurethan foam design. The research is well designed. The results are presented clearly.

Basing on the results it is possible to create different foam of different hydrodynamic behaviour.

In my opinion, the paper can be published as is.

Author Response

Dear Reviewer, 

Thank you for your time and positive comments. The final manuscript has some minor changes in accordance to other reviewers comments.

Kind Regards,